# Insulin sensitivity is preserved in mice made obese by feeding a high starch diet

Amanda E Brandon[1,2]*[†], Lewin Small[2†], Tuong-Vi Nguyen[2], Eurwin Suryana[2], Henry Gong[1], Christian Yassmin[1], Sarah E Hancock[3], Tamara Pulpitel[4], Sophie Stonehouse[4], Letisha Prescott[4], Melkam A Kebede[4], Belinda Yau[4], Lake-Ee Quek[5], Greg M Kowalski[6,7], Clinton R Bruce[6], Nigel Turner[3], Gregory J Cooney[1,2]

[1]School of Medical Sciences, University of Sydney, Sydney, Australia; [2]Diabetes and Metabolism Division, Garvan Institute of Medical Research, Sydney, Australia; [3]Department of Pharmacology, School of Medical Sciences, University of New South Wales, Sydney, Australia; [4]School of Life and Environmental Sciences, Charles Perkins Centre, University of Sydney, Sydney, Australia; [5]School of Mathematics and Statistics, Charles Perkins Centre, University of Sydney, Sydney, Australia; [6]Institute for Physical Activity and Nutrition, School of Exercise and Nutrition Sciences, Deakin University, Melbourne, Australia; [7]Metabolic Research Unit, School of Medicine, Deakin University, Melbourne, Australia

*For correspondence:
amanda.brandon@sydney.edu.au

[†]These authors contributed equally to this work

Competing interest: The authors declare that no competing interests exist.

**Abstract** Obesity is generally associated with insulin resistance in liver and muscle and increased risk of developing type 2 diabetes, however there is a population of obese people that remain insulin sensitive. Similarly, recent work suggests that mice fed high carbohydrate diets can become obese without apparent glucose intolerance. To investigate this phenomenon further, we fed mice either a high fat (Hi-F) or high starch (Hi-ST) diet and measured adiposity, glucose tolerance, insulin sensitivity, and tissue lipids compared to control mice fed a standard laboratory chow. Both Hi-ST and Hi-F mice accumulated a similar amount of fat and tissue triglyceride compared to chow-fed mice. However, while Hi-F diet mice developed glucose intolerance as well as liver and muscle insulin resistance (assessed via euglycaemic/hyperinsulinaemic clamp), obese Hi-ST mice maintained glucose tolerance and insulin action similar to lean, chow-fed controls. This preservation of insulin action despite obesity in Hi-ST mice was associated with differences in de novo lipogenesis and levels of C22:0 ceramide in liver and C18:0 ceramide in muscle. This indicates that dietary manipulation can influence insulin action independently of the level of adiposity and that the presence of specific ceramide species correlates with these differences.

## Editor's evaluation

This important study evaluates the effects of two distinct dietary methods that cause obesity in mice (high fat vs high starch) on insulin sensitivity and glucose homeostasis. The authors present compelling data showing that high starch feeding causes obesity without deleterious effects on insulin sensitivity. This work will have broad impact in the field and will help define the lipid mediators of metabolic disease.

## Introduction

Obesity is a major risk factor for the development of dysregulated glucose metabolism, insulin resistance, and type 2 diabetes in humans (*Lin and Li, 2021*; *Muoio and Newgard, 2006*). However, the

degree of insulin resistance does not always correlate with the extent of obesity (*Gan et al., 2003*) and there exists a population of individuals who are obese yet exhibit no difference in insulin action compared to groups of lean individuals (*Tonks et al., 2016*), and lean individuals who are metabolically unhealthy (*Stefan et al., 2017*). These reports of obese individuals with preserved insulin action have led to the recognition of a population described as metabolically healthy obese (MHO) but the mechanisms behind the different metabolic profiles in equally obese individuals remain unclear. It has been suggested that MHO individuals are simply on a different trajectory to more overt metabolic dysfunction but there is also the possibility that genetic background, body fat distribution, or diet and exercise history might influence the relationship between obesity and insulin action (*Vilela et al., 2021*; *Gómez-Zorita et al., 2021*).

Animal models used to investigate the mechanistic relationship between obesity and metabolic dysfunction normally employ a Hi-F diet regime (45–60% of total calories) to produce obesity and insulin resistance (reviewed in *Small et al., 2018*). As with humans, the level of insulin resistance observed in animal models does not necessarily correlate with the accumulation of fat in diet-induced obesity. It has been observed in mice and rats that insulin action measured by hyperinsulinaemic/euglycaemic clamp or glucose tolerance tests is decreased after 2 or 3 weeks of feeding a Hi-F diet but does not significantly worsen over the subsequent 10–20 weeks even though the animals continue to accumulate fat and become obese (*Turner et al., 2013*; *Turner et al., 2007*; *Burchfield et al., 2018*). There are also data showing that some strains of mice can become obese eating a Hi-F diet without deleterious effects on glucose tolerance (*Montgomery et al., 2013a*; *Stöckli et al., 2017*; *Appiakannan et al., 2020*). Clearly genetic makeup can impact the relationship between obesity and glucoregulation, but less is known about the relative importance of diet composition and total fat mass in the regulation of insulin action at the whole body and tissue level. In a study assessing the impact of diet composition on longevity and metabolic health in mice, animals fed diets lower in protein and fat and higher in carbohydrate had the greatest median lifespan (*Solon-Biet et al., 2014*). Interestingly, mice on these lower protein, higher carbohydrate diets also exhibited better measures of glucose tolerance despite accumulating similar amounts of adipose tissue to mice on Hi-F diets that exhibited poor glucose tolerance (*Solon-Biet et al., 2014*).

To further explore the relationship between diet composition, obesity, and insulin action in mice, we have examined body fat, glucose tolerance, and insulin action by hyperinsulinaemic/euglycaemic clamp in animals fed diets high in carbohydrate (starch) or high in fat (lard) and compared outcomes to mice fed standard laboratory chow. The results show that high-starch (Hi-ST) and Hi-F fed mice become equally obese compared to chow fed mice, but that only fat fed mice develop a significant degree of glucose intolerance and insulin resistance. These findings suggest that, at least in mice, the composition of the current diet is a greater determinant of glucose homeostasis than the degree of adiposity.

## Results

### Body weight, composition, and whole body calorimetry

Mice fed a Hi-ST diet had a similar body weight trajectory to that of mice fed a Hi-F diet and by the end of the feeding period, both these groups were heavier than the chow controls and there was no significant difference in body weight between Hi-ST and Hi-F fed mice (*Figure 1A*). Body composition measured at 4, 6, and 12 weeks showed that this increase in body weight was due to an increase in fat mass as lean mass was similar between the three groups (*Figure 1B and C*). Mice on the two dietary interventions consumed more energy per day compared to the control chow-fed group (*Figure 1D*). Energy expenditure (EE) was similar between groups (*Figure 1E and F*) with mice displaying a positive relationship between body mass and EE regardless of diet (*Figure 1G*). Ambulatory activity was not different between groups (*Figure 1H*). Hi-F fed mice displayed a lower respiratory quotient (*Figure 1I and J*), indicating a higher ratio of lipid to carbohydrate oxidation, while Hi-ST mice showed an intermediate respiratory quotient between Hi-F mice and chow control mice possibly due to a higher dietary fat content of the Hi-ST diet compared to chow (calorically 21% vs. 6%, respectively).

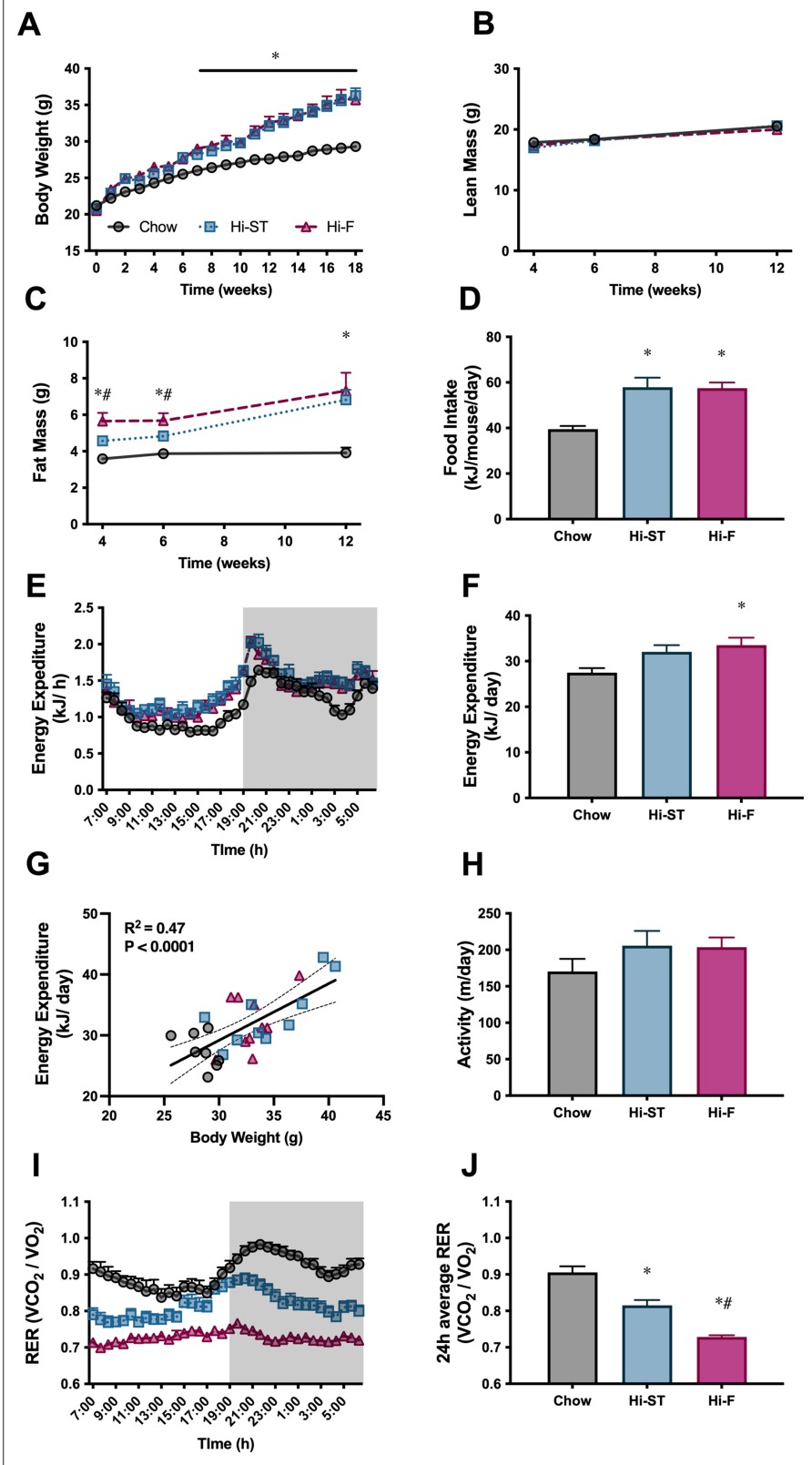

**Figure 1.** High starch (Hi-ST) and high fat (Hi-F) similarly increase body weight, fat mass, and food intake without changes in lean mass or energy expenditure. (**A**) Body weight, n=18–20, (**B**) lean mass, (**C**) fat mass, n=8–10, (**D**) food intake, n=4 (averaged within cages), (**E**) energy expenditure, (**F**) 24 hr average of energy expenditure, (**G**) relationship between energy expenditure and body weight, (**H**) ambulatory activity, (**I**) respiratory exchange

*Figure 1 continued on next page*

*Figure 1 continued*

ratio (RER), (**J**) 24 hr average of RER, n=8–10. Grey – chow, blue – Hi-ST, pink – Hi-F. Data presented as mean ± SEM. Data analysed with a one-way ANOVA or two-way repeated measures ANOVA where appropriate. Panel G analysed by linear regression. *p<0.05 different to chow; #p<0.05 different to Hi-ST.

The online version of this article includes the following source data for figure 1:

**Source data 1.** Raw data for *Figure 1*.

## Whole body glucose homeostasis and insulin secretion

Oral glucose tolerance tests (oGTT) were performed at 4 weeks (*Figure 2A and B*) and 12 weeks (*Figure 2D and E*) after the start of the dietary intervention. While the Hi-F diet caused the expected decrease in glucose tolerance at both time points, glucose tolerance in the Hi-ST mice, remained very similar to chow control mice, even though Hi-ST mice had similar levels of adiposity as the Hi-F mice. After both 4 and 12 weeks of diet, Hi-ST and Hi-F mice displayed a higher fasting and glucose-stimulated blood insulin concentration during the GTT compared to chow controls with Hi-ST mice returning to basal levels faster than the Hi-F group (*Figure 2C and F*).

To gain additional insight into glucose handling kinetics during the oGTT, 14 weeks after dietary intervention, an additional cohort of mice underwent a stable isotope labelled oGTT (6,6-$^2$H-glucose). The glucose excursions showed a similar pattern to that seen previously, with Hi-ST mice having improved glucose tolerance compared to the Hi-F mice (*Figure 2G*). While the plasma 6,6-$^2$H-glucose enrichment curves were similar in all three groups (*Figure 2H*), there was a significant and consistent difference in the exogenous glucose concentration curves, with Hi-F being higher than chow and Hi-ST (*Figure 2I*), thus indicating that Hi-ST mice had improved whole body glucose disposal during the oGTT when compared to Hi-F. When endogenous glucose levels were expressed as a change from baseline, there were only small changes evident between Hi-ST and Hi-F at 15 and 30 min post glucose load, suggesting that Hi-ST mice may have also had a brief period of enhanced endogenous glucose production (EGP) suppression during the early period (0–30 min) of the OGTT compared to Hi-F mice.

Since both the Hi-F and the Hi-ST animals exhibited higher insulin levels during the oGTT, glucose-stimulated insulin secretion was investigated in isolated islets from the different dietary groups to see if there were any differences. The amount of insulin released from isolated islets in response to 16.7 mM glucose was similar in all groups (*Figure 2K*), however there was an increase in insulin content that was only significant in the Hi-F group (*Figure 2L*). Thus, it would appear that inherent differences in islet function are not the driver of glucose tolerance differences observed in the Hi-ST fed and Hi-F fed mice.

## Euglycaemic-hyperinsulinaemic clamp

To investigate if the maintenance of glucose tolerance in Hi-ST mice was associated with preservation of insulin sensitivity, euglycaemic-hyperinsulinaemic clamps were performed. Blood glucose was maintained at euglycaemia (~8 mM) throughout the clamp (*Figure 3A*) and plasma insulin levels were elevated to a similar extent by the insulin infusion between groups (*Figure 3B*). All three groups had similar basal NEFA (chow 0.73±0.07; Hi-ST 0.78±0.08; Hi-F 0.67±0.09 mM) and were able to suppress NEFA levels to a similar extent (chow 0.18±0.03; Hi-ST 0.22±0.03; Hi-F 0.29±0.05 mM). The glucose infusion rate (GIR) needed to maintain euglycaemia was lower in the Hi-F animals compared to both the chow and Hi-ST, indicating a reduced whole body insulin action (*Figure 3C and D*; p>0.05), while there was no significant difference in GIR between the obese Hi-ST and lean chow control mice. Insulin infusion increased glucose disposal in all three groups and while the insulin-stimulated glucose disposal in Hi-F mice was lower than chow or Hi-ST mice, the difference did not reach statistical significance (p=0.079). Although there was no difference in the basal hepatic glucose production, the ability of insulin to suppress hepatic glucose output (HGO) was impaired in the Hi-F animals when compared to both the chow and Hi-ST groups (*Figure 3F and G*). Tissue glucose uptake data clearly demonstrated that there was preservation of skeletal and heart muscle insulin action in the Hi-ST mice, while the Hi-F showed reduced insulin action indicative of insulin resistance (*Figure 3H*). White adipose tissue (WAT) however was insulin resistant in both the Hi-ST and Hi-F groups compared to controls (*Figure 3I*). On the other hand, brown adipose tissue (BAT) glucose uptake was fully preserved in

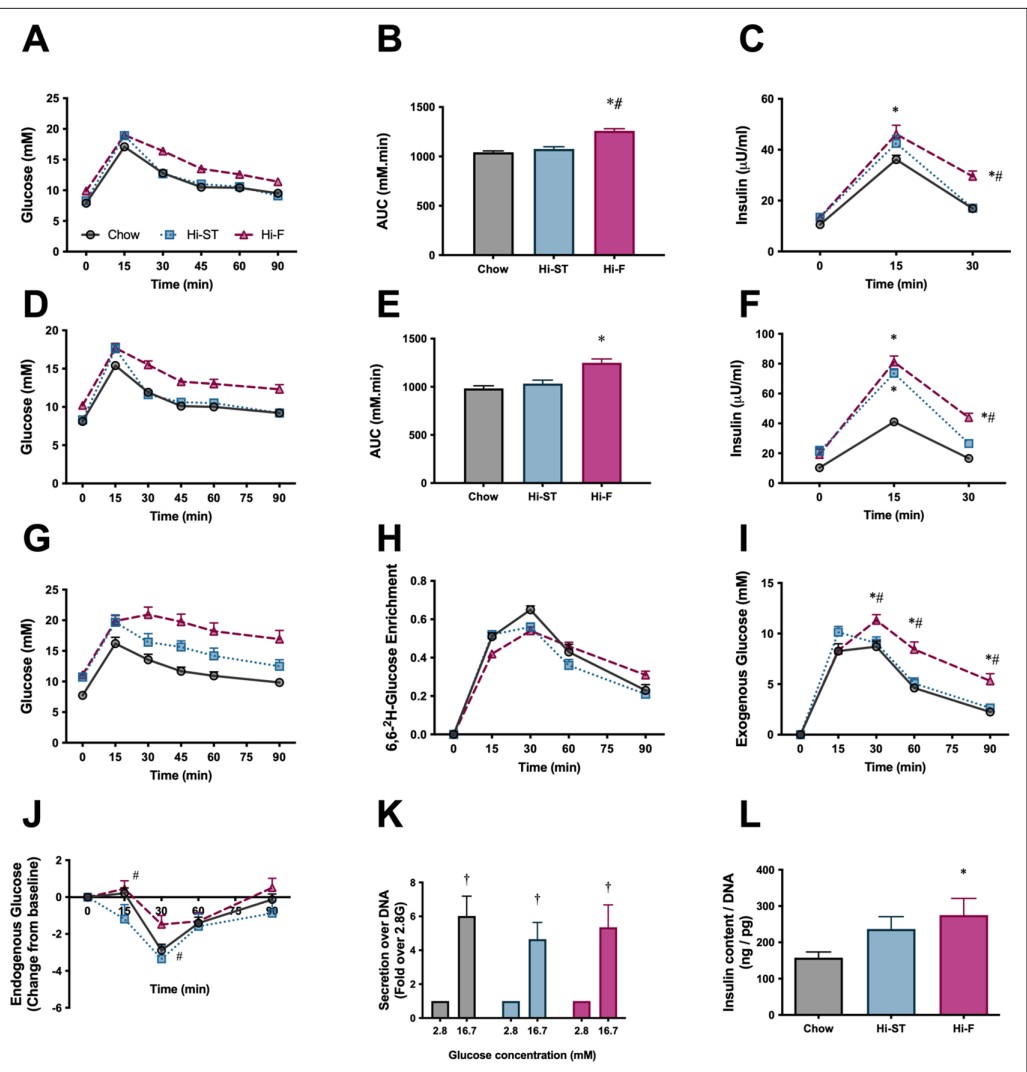

**Figure 2.** Glucose tolerance of high starch (Hi-ST) mice is similar to chow despite increased adiposity and may be due to liver, but not β-cell, involvement. (**A**) Oral glucose tolerance test (oGTT) after 4 weeks of feeding, (**B**) area under the curve for the oGTT after 4 weeks of feeding, n=45, (**C**) insulin released during the oGTT after 4 weeks of feeding, n=42–45. (**D**) oGTT after 12 weeks of feeding, (**E**) area under the curve for the oGTT after 12 weeks of feeding, (**F**) insulin released during the oGTT after 12 weeks of feeding, n=19–20. (**G**) oGTT with deuterated glucose after 14 weeks of feeding, n=10, (**H**) deuterated glucose enrichment curves, (**I**) endogenous glucose production during the oGTT, (**J**) endogenous glucose production as expressed as a change from baseline, n=10. (**K**) Glucose-stimulated insulin secretion, n=5–6, (**L**) insulin content of β-cell (53–60 islets). Grey – chow, blue – Hi-ST, pink – high fat (Hi-F). Data presented as mean ± SEM. Data analysed with a one-way ANOVA or two-way repeated measures ANOVA where appropriate. *p<0.05 different to chow; #p<0.05 different to Hi-ST; ‡p<0.05 different to 2.8 of the same dietary group.

The online version of this article includes the following source data for figure 2:

**Source data 1.** Raw data for *Figure 2*.

the Hi-ST group while being significantly reduced in the Hi-F fed animals (*Figure 3I*). The differences in insulin sensitivity of liver or quadriceps muscle were not associated with significant differences in the phosphorylation status of Akt (*Figure 3J and K*, respectively). There were also no differences in the phosphorylation status of Akt in either the epididymal or subcutaneous WAT (*Figure 3L and M*, respectively).

Mitochondrial enzyme activities in the liver showed an increase in citrate synthase (CS) in the Hi-ST group while β-hydroxyacyl CoA dehydrogenase (βHAD) was increased in the Hi-F animals (*Table 1*).

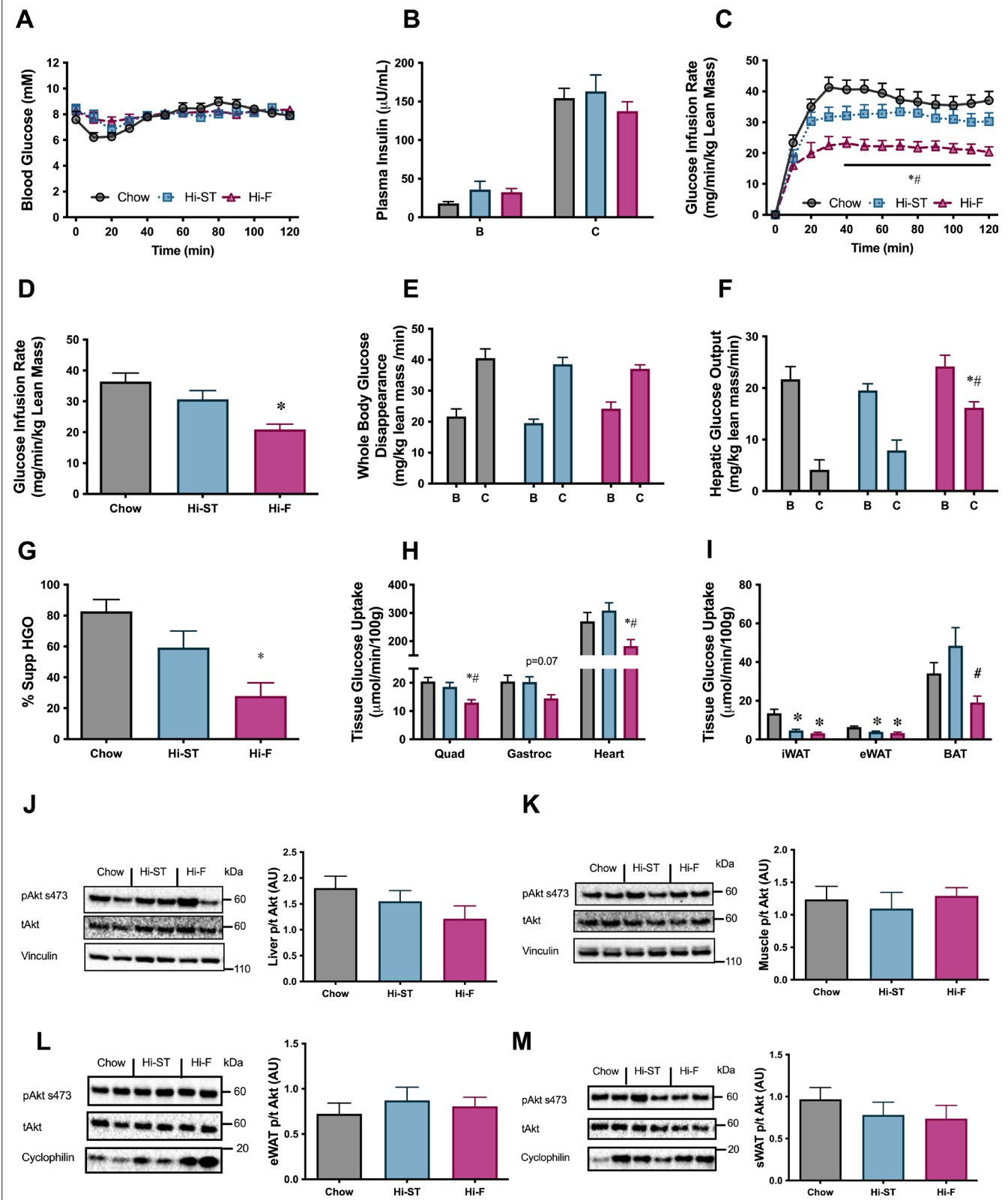

**Figure 3.** Euglycaemic-hyperinsulinaemic clamp parameters show preservation of whole body, liver and skeletal muscle insulin resistance. (**A**) Blood glucose, (**B**) plasma insulin, (**C**) glucose infusion rate, (**D**) average glucose infusion rate of 90–120 min, (**E**) whole body glucose disappearance, (**F**) hepatic glucose output (HGO), (**G**) percent suppression of HGO, (**H**) muscle glucose uptake, (**I**) adipose tissue glucose uptake, (**J**) phosphorylation of Akt in liver after the clamp, (**K**) phosphorylation of Akt in quadriceps muscle after the clamp, (**L**) phosphorylation of Akt in epididymal white adipose tissue (eWAT)

*Figure 3 continued on next page*

*Figure 3 continued*

after the clamp, (**M**) phosphorylation of Akt in subcutaneous white adipose tissue (sWAT) after the clamp. B, basal state; C, clamped state. Grey – chow, blue – high starch (Hi-ST), pink – high fat (Hi-F). Data presented as mean ± SEM. Data analysed with a one-way ANOVA or two-way repeated measures ANOVA where appropriate. n=8–11. *p<0.05 different to chow; #p<0.05 different to Hi-ST.

The online version of this article includes the following source data for figure 3:

**Source data 1.** Raw data for *Figure 3*.

**Source data 2.** Raw western blot images for *Figure 3*.

In muscle there were increases in CS and βHAD in both the Hi-ST and Hi-F groups (*Table 1*). The Hi-F animals also had a decrease in the activity of pyruvate dehydrogenase (PDH) in muscle but a lower activity of PDH in liver of Hi-F mice did not reach statistical significance. Metabolomic analysis of glycolytic and TCA cycle intermediates after insulin stimulation during the clamp showed very little change across the three dietary groups in both muscle (*Supplementary file 1a*) and liver (*Supplementary file 1b*).

Taken together, this data shows that preservation of insulin action at the whole body level in animals fed a Hi-ST diet is paralleled by a maintenance of insulin action at the tissue level (liver, skeletal muscle, BAT) and this was associated with altered activity of enzymes responsible for the direct oxidation of glucose and fatty acids in muscle and liver, rather than alterations in Akt phosphorylation. Additionally, this data suggests that in diet-induced obese mice, hepatic insulin resistance seems to be the largest contributing factor to the lower GIR during a clamp, while during a GTT, hepatic and peripheral insulin resistance seems to make similar contributions to the relative glucose intolerance observed in Hi-F mice (*Figure 2I and J*).

## Tissue parameters

Consistent with the increase in whole body fat mass in the Hi-ST and Hi-F groups, as determined by EchoMRI (*Figure 1C*), we saw an increase in the weights of three WAT depots (*Figure 4A*; epididymal [Epi], subcutaneous [SC], and retroperitoneal [RP]). Liver (control 1.1±0.3; Hi-ST 1.2±0.1; Hi-F 1.2±0.1 g) and BAT (*Figure 4A*) mass did not change with diet treatment. Adipocyte size in both the epididymal and subcutaneous fat pads were, on average, increased in animals fed Hi-ST and Hi-F when compared to chow (*Figure 4B and C*). As expected with the increase in fat mass, plasma leptin levels increased in the Hi-ST and Hi-F animals (*Figure 4D*). Plasma adiponectin levels were not different between the groups (*Figure 4E*). Interestingly, total triglyceride levels in both muscle and

**Table 1.** Enzyme activities in liver and muscle.

| Enzyme activity<br>µmol/min/g protein | Chow | Hi-ST | Hi-F |
|---|---|---|---|
| Liver | | | |
| Citrate synthase | 110.8±7.3 | 140.1±3.0* | 119.9±6.1 |
| βHAD | 100.4±5.2 | 117.0±2.1 | 122.7±7.0* |
| Succinate dehydrogenase | 14.1±0.6 | 15.1±0.4 | 15.1±1.2 |
| Pyruvate dehydrogenase | 0.90±0.21 | 0.73±0.33 | 0.36±0.07 |
| Quadriceps muscle | | | |
| Citrate synthase | 188.7±5.1 | 208.0±6.0* | 223.1±3.7* |
| βHAD | 23.9±1.2 | 28.4±1.4* | 35.4±1.2*# |
| Succinate dehydrogenase | 2.7±0.1 | 2.7±0.1 | 2.5±0.2 |
| Pyruvate dehydrogenase | 0.46±0.08 | 0.33±0.10 | 0.15±0.04* |

Data expressed as mean ± SEM. n=7–10. *p<0.05 compared to chow; #p<0.05 compared to Hi-ST.

The online version of this article includes the following source data for table 1:

**Source data 1.** Raw data for *Table 1*.

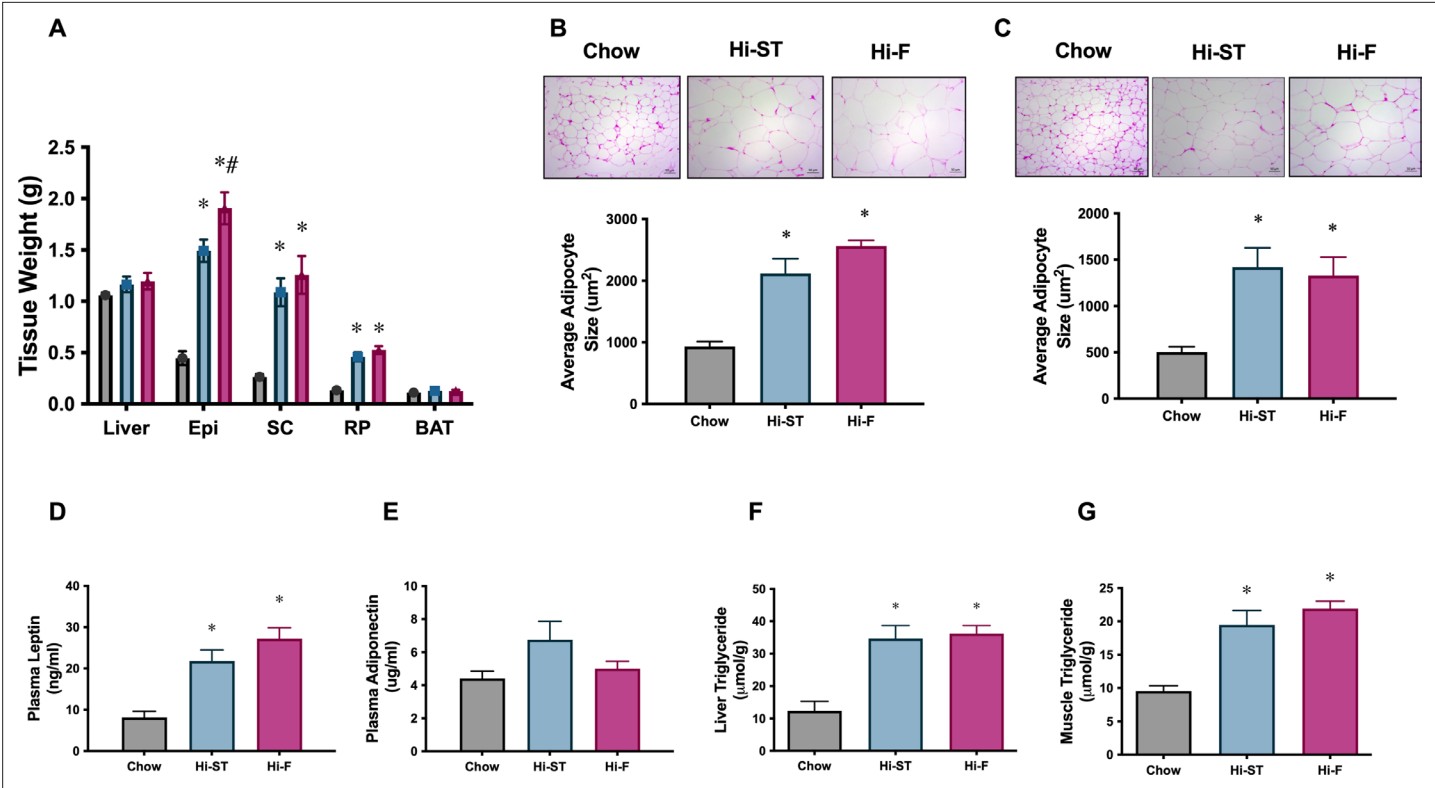

**Figure 4.** Adipose tissue weights, and adipocyte size, were increased to a similar extent in high starch (Hi-ST) and high fat (Hi-F) as were plasma leptin and the level of triglycerides in liver and muscle. (**A**) Tissue weights, n=9–11, (**B**) average adipocyte size in epididymal white adipose tissue (WAT), (**C**) average adipocyte size in subcutaneous WAT, n=3–5, plasma levels of (**D**) leptin and (**E**) adiponectin, n=10–11. (**F**) Liver triglyceride levels, n=5–8, (**G**) quadriceps muscle triglyceride levels, n=9–12. Grey – chow, blue – Hi-ST, pink – Hi-F. Data presented as mean ± SEM. Data analysed with a one-way ANOVA. *p<0.05 different to chow; #p<0.05 different to Hi-ST.

The online version of this article includes the following source data for figure 4:

**Source data 1.** Raw data for *Figure 4*.

liver showed a similar increase in the Hi-F and Hi-ST animals (*Figure 4F and G*), despite the differences in glucose tolerance and insulin sensitivity.

To investigate the contribution of de novo lipogenesis (DNL) to the tissue triglyceride pool, a separate cohort of mice were fed for 4 weeks, received an IP injection of $^3H_2O$ in the postprandial state and the incorporation of $^3H$ into newly synthesised lipid was measured. Four weeks of feeding was sufficient to increase total triglyceride levels in the liver of Hi-F and Hi-ST animals compared to control (*Figure 5A*). This increase in liver triglyceride was associated with a suppression of liver DNL in the Hi-F animals compared to controls, with an increase in DNL seen in the Hi-ST animals (*Figure 5B*). There was no difference between the groups in the rates of DNL in BAT or WAT (*Figure 5C and D*). This same pattern was also seen in the abundance of key lipogenic enzymes acetyl CoA carboxylase (ACC), fatty acid synthase (FAS), and stearoyl-CoA desaturase-1 (SCD-1), which were increased in the Hi-ST animals and decreased in the Hi-F groups compared to chow controls (*Figure 4E and F*). Activities of two other enzymes, ATP citrate lysase (ACL), involved in the production of cytosolic acetyl-CoA and glucose-6-phosphate dehydrogenase (G6PDH), which generates NADPH that is used as a cofactor for lipogenesis, also mirrored these results (*Figure 5G and H*).

This data indicates that liver is the primary site of DNL in the mouse since the rate of DNL in this organ was much higher than in any of the WAT depots. The data also indicated that DNL is lower in the Hi-F animals that have abundant access to fat from the diet, and higher in the Hi-ST animals where fat accumulation depends largely on DNL conversion of dietary carbohydrate to fat.

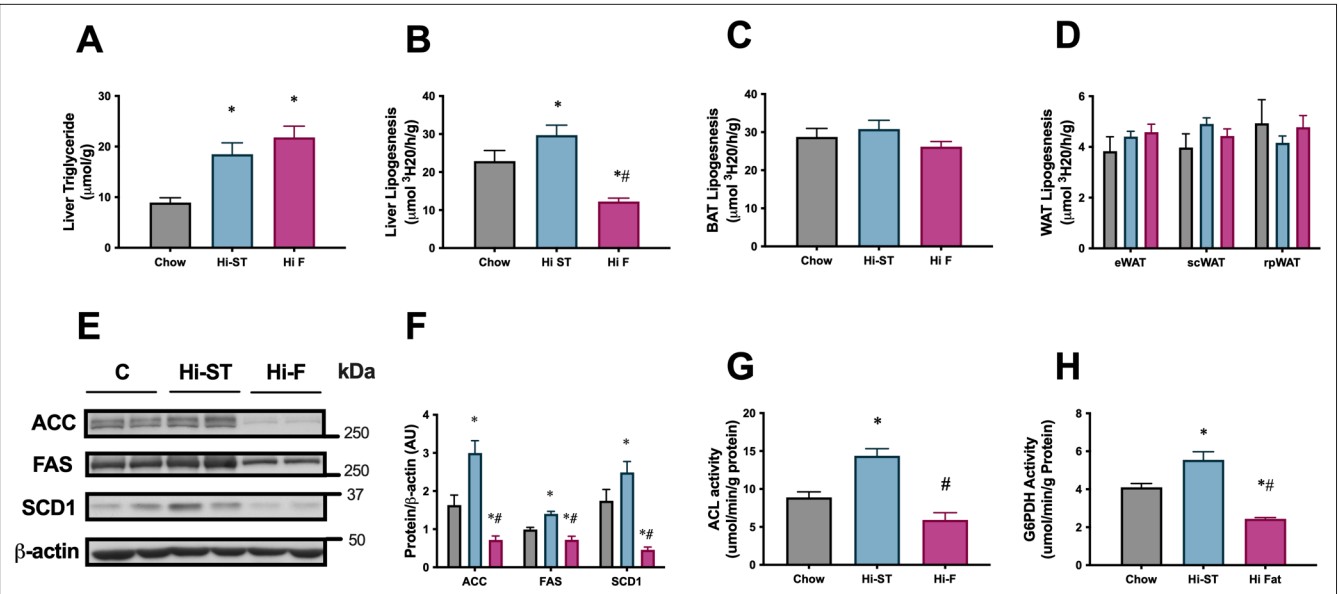

**Figure 5.** The increased liver triglycerides in the high starch (Hi-ST) mice was due to an upregulation of the de novo lipogenic pathway. (**A**) Liver triglycerides, (**B**) liver lipogenic rate, (**C**) brown adipose tissue (BAT) lipogenic rate, (**D**) white adipose tissue (WAT) lipogenic rates, n=12–15, (**E**) western blots of acetyl CoA carboxylase (ACC), fatty acid synthase (FAS), and stearoyl-CoA desaturase-1 (SCD-1), (**F**) densitometry of protein levels of ACC, FAS, and SCD-1, (**G**) ATP citrate lysase (ACL) activity and (**H**) glucose-6-phosphate dehydrogenase (G6PDH) activity, n=9–10. Grey – chow, blue – Hi-ST, pink – high fat (Hi-F). Data presented as mean ± SEM. Data analysed with a one-way ANOVA. *p<0.05 different to chow; #p<0.05 different to Hi-ST.

The online version of this article includes the following source data for figure 5:

**Source data 1.** Raw data for *Figure 5*.

**Source data 2.** Raw western blot images for *Figure 5*.

## Lipidomic analysis

Since triglyceride stores are considered a relatively benign form of lipid, we used lipidomic analysis to assess other, more biologically active lipid species, in particular diacylglycerols (DAGs) and ceramides (Cer). In both the liver and the muscle, this lipidomic analysis confirmed an increase in total triglyceride content that was similar in both Hi-ST and Hi-F groups (*Figure 6A and E*). Total DAG levels were similarly increased in both dietary groups, in both tissues (*Figure 6A and E*), and not surprisingly, this was reflected in the amounts of the more abundant DAG species (*Figure 6B and F*). In the muscle there were no other significant changes in total levels of other lipid species (sphingomyelin [SM], phosphatidylcholine [PC], phosphatidylethanolamine [PE], or phosphatidylserine [PS]; *Figure 6A*), although there were some changes in specific species (*Figure 6—figure supplement 1*). In the liver there were increases in total levels of PE and cholesterol esters (CE; *Figure 6E*). There was no clear pattern to changes in PE species, but almost all the CE species were greater in the dietary intervention groups compared to chow, with six out of the nine species being increased to a greater extent in the Hi-ST group (*Figure 6—figure supplement 2*). This increase in CE is not surprising in the Hi-ST animals since the same precursor, acetyl-CoA, is used in both DNL and CE synthesis.

Ceramide levels in both tissues showed interesting differences that correlated with tissue-specific measures of insulin action. In quadriceps muscle, Cer18:0 was the predominant lipid species and while it was slightly increased in the Hi-ST animals compared to chow controls, there was a larger increase in the Hi-F group (*Figure 6C*). The Cer18:0 was also inversely correlated with glucose uptake measured in the same muscle (*Figure 6D*). In the liver, Cer22:0, Cer24:0, and Cer24:1 were the predominant ceramide species. Of these abundant species, Cer22:0 was the only one that significantly changed with diet and demonstrated the same pattern as observed for liver insulin action (i.e. suppression of HGO during the clamp). Specifically, Cer22:0 levels were considerably higher in the Hi-F compared to chows with the Hi-ST being intermediate between the two (*Figure 6G*). This species of ceramide was negatively correlated with suppression of HGO, an index of liver insulin action (*Figure 6H*). Cer20:0,

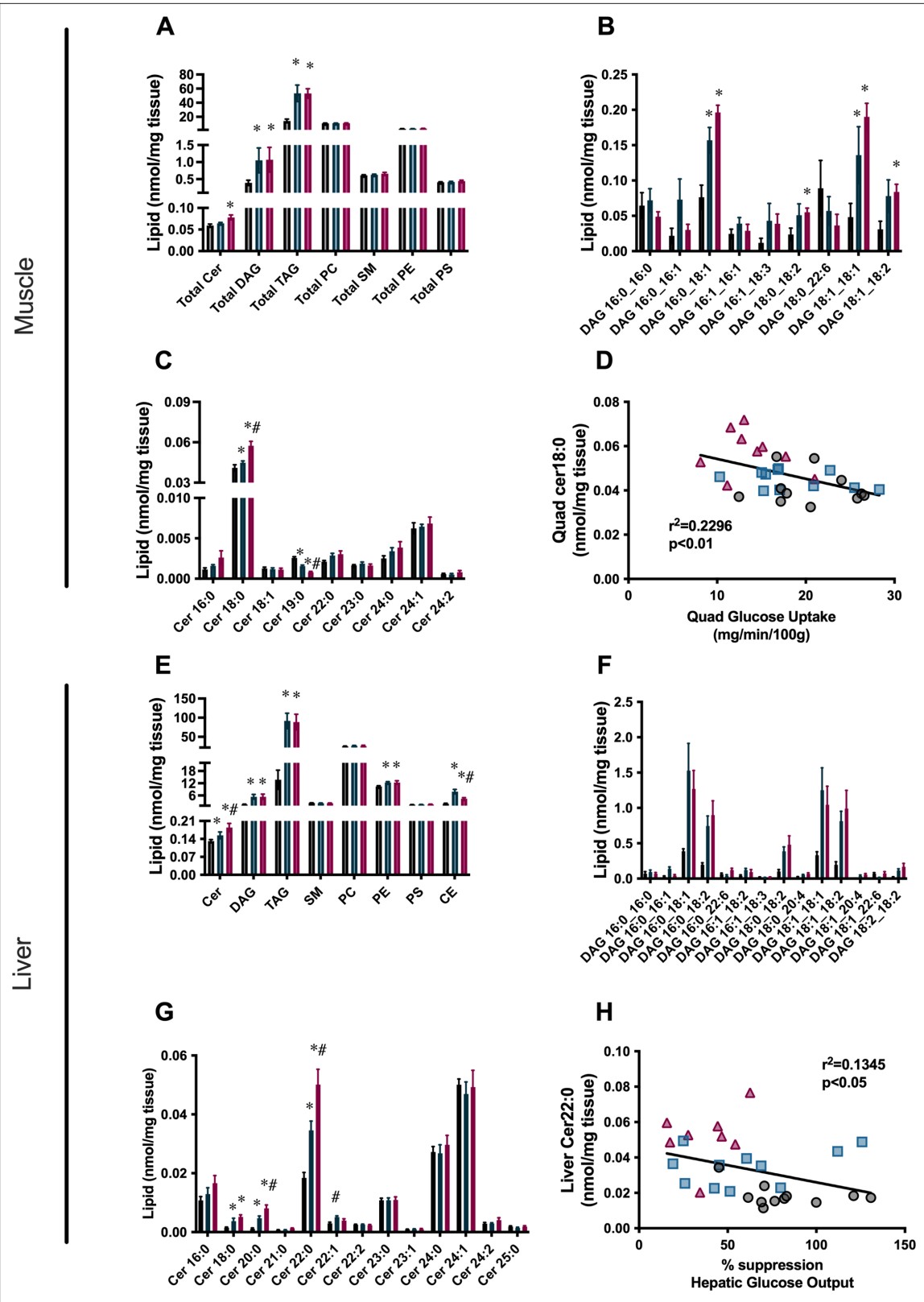

**Figure 6.** Reduction in total ceramide levels, as well as specific ceramide species that are correlated with insulin sensitivity, is found in the liver and quadriceps muscle of high starch (Hi-ST) fed mice. (**A**) Total lipid levels, (**B**) diacylglycerol (DAG) species, (**C**) ceramide species, and (**D**) the relationship between insulin sensitivity and Cer18:0 levels in muscle. (**E**) Total lipid levels, (**F**) DAG species, (**G**) ceramide species, and (**H**) the relationship between insulin sensitivity and Cer22:0 levels in liver. Grey – chow, blue – Hi-ST, pink – high fat (Hi-F). Data presented as mean ± SEM. Data analysed with a

*Figure 6 continued on next page*

*Figure 6 continued*

Kruskal-Wallis test with pairwise Wilcoxon rank sum used for post hoc comparisons. Data were corrected for false discovery rate using the method of Benjamini-Hochberg. n=8–11. *p<0.05 different to chow; #p<0.05 different to Hi-ST.

The online version of this article includes the following source data and figure supplement(s) for figure 6:

**Source data 1.** Raw data for *Figure 6*.

**Figure supplement 1.** Levels of sphingomyelin (SM), phosphatidylcholine (PC), phosphatidylethanolamine (PE), or phosphatidylserine (PS) in muscle.

**Figure supplement 1—source data 1.** Raw data for *Figure 6—figure supplement 1*.

**Figure supplement 2.** Levels of sphingomyelin (SM), phosphatidylcholine (PC), phosphatidylethanolamine (PE), phosphatidylserine (PS), and cholesterol esters (CE) in liver.

**Figure supplement 2—source data 1.** Raw data for *Figure 6—figure supplement 2*.

while lower in total levels, showed a similar pattern of abundance (*Figure 6G*) and negative correlation with insulin action ($r^2$=0.139, p<0.05).

## Discussion

The relationship between body fat content and insulin action is usually considered to be a negative one, where increasing body fat is associated with a reduction in insulin action (*Gan et al., 2003*; *Owei et al., 2017*). However, there have now been several reports of individuals with significant obesity but little, or no, difference in metabolic parameters such as insulin resistance when compared to a lean healthy group of individuals (*Tonks et al., 2016*). What makes these obese individuals apparently protected from metabolic disease remains unclear and although there are some mouse models that exhibit a similar obese, insulin-sensitive phenotype these have been produced by genetic manipulations (*Hotamisligil et al., 1996*; *Kim et al., 2007*). Animal models used to investigate the relationship between obesity and insulin action often use a diet high in fat and high in simple carbohydrates (sucrose, fructose) to cause fat gain, insulin resistance, and hepatic steatosis (*Kim et al., 1999*; *Malik et al., 2010*). Dietary manipulation models in mice may have more relevance to humans than monogenic models of obesity as overconsumption of calories invariably leads to an increase in fat mass in both mice and humans. The current study reports on a dietary intervention that produces significant obesity but maintains normal insulin action in mice providing some mechanistic understanding of the phenomenon of obesity with preserved insulin sensitivity.

Mice fed a Hi-ST diet were found to have a similar (but not identical) glucose tolerance and insulin sensitivity compared to lean chow controls, even though they accumulated the same amount of fat mass as mice fed a Hi-F diet. As mentioned above, an increased amount of adipose tissue is usually associated with poor glucose tolerance, decreased insulin action, and detrimental metabolic outcomes. Thus the differential glucose homeostasis and insulin action but similar increased fat mass of the Hi-ST and Hi-F mice provides a useful dichotomy to investigate any mechanistic relationship between diet, fat mass, and insulin action. Our results suggest that adipocytes are unlikely to mediate the maintenance of whole body sensitivity in Hi-ST mice as absolute mass, adipocyte size (i.e. cross-sectional area), and adipose tissue depot insulin sensitivity were similar between Hi-ST and Hi-F fed mice. Therefore, the tissues responsible for the bulk of whole body glucose metabolism, liver and muscle, were more likely to exhibit differences that contribute to our understanding of the maintenance of clamp insulin sensitivity in the presence of obesity in Hi-ST fed mice. Glucose tolerance was also similar in chow and Hi-ST fed mice despite the obesity in Hi-ST mice. However, there were differences in basal insulinaemia and insulin response during the glucose tolerance test that indicate that there were metabolic differences between chow and Hi-ST mice even though measures of glucose tolerance and clamp insulin sensitivity were similar. Interestingly, despite the preservation of clamp insulin sensitivity in Hi-ST mice, both liver and muscle of these mice had equivalent amounts of triglyceride storage as observed in Hi-F mice (*Figure 4*). Therefore, differences in ectopic triglyceride levels in liver and muscle cannot explain the divergent insulin action in Hi-ST and Hi-F mice. Further investigations into the source of the greater fat deposition in Hi-ST and Hi-F mice showed that Hi-ST animals had an increased rate of DNL in liver and increased levels of enzymes involved in the DNL pathway, while in the Hi-F group, levels of these enzymes were decreased compared to chow and Hi-ST fed mice (*Figure 5*). This suggests that upregulation of pathways converting carbohydrate to

lipid could underpin the ability of Hi-ST mice to clear glucose and maintain insulin sensitivity. This data indicates that synthesising fat for storage from dietary carbohydrate is less detrimental to insulin action and glucose homeostasis than metabolising and storing fat from the diet. It seems mechanistically plausible that an animal consuming mainly carbohydrate would maintain or increase capacity for utilising this nutrient even if the end result is storage of triglyceride, while an animal consuming mainly fat would increase capacity for utilising fat and downregulate carbohydrate metabolic pathways.

Since triglycerides are thought to be a metabolically benign form of lipid, we performed targeted lipidomic analysis on muscle and liver of the clamped mice to investigate changes in other lipid species. Differences in lipid species were detected between Hi-F and Hi-ST, particularly in ceramides, that have been shown previously to correlate with insulin sensitivity/resistance (*Tonks et al., 2016*; *Turpin-Nolan et al., 2019*; *Bergman et al., 2016*). In skeletal muscle, Cer18:0 was increased in the Hi-F group compared to chow and Hi-ST animals. This is similar to the observation in humans where obese, insulin-sensitive individuals have similar levels of Cer18:0 to lean individuals, while obese, insulin-resistant individuals had increased levels (*Tonks et al., 2016*). Previous studies have also shown Cer18:0 content in muscle to be increased in mice fed a Hi-F diet (*Turner et al., 2013*; *Turpin-Nolan et al., 2019*; *Frangioudakis et al., 2010*) while reductions of Cer18:0, through deletion of ceramide synthase 1, has been associated with improved glucose tolerance and insulin sensitivity (*Turpin-Nolan et al., 2019*; *Błachnio-Zabielska et al., 2022*). Our data also showed a negative correlation between skeletal muscle insulin sensitivity (insulin-stimulated glucose uptake) and muscle Cer18:0 levels. A similar relationship has previously been demonstrated in humans (*Bergman et al., 2016*; *Perreault et al., 2018*). However, this relationship has not always been found (*Turner et al., 2018*). DAGs are postulated to be mediators of insulin resistance, providing a link between increased lipid and reduced insulin-stimulated glucose uptake (*Birkenfeld and Shulman, 2014*). DAG content was increased in the Hi-F and the Hi-ST muscle to a similar extent and therefore cannot explain the differential insulin action observed between Hi-ST and Hi-F mice. Taken together, our data suggests that increased Cer18:0 is more closely linked to the development of (Hi-F) skeletal muscle insulin resistance. Mechanistically, it has been suggested that changes in ceramide levels reduce insulin action by altering insulin signalling (*Summers et al., 2019*). Interestingly, we observed no difference in insulin-stimulated phosphorylation of Akt in all three dietary groups suggesting that the mechanism relating changes in ceramide to reduced insulin-stimulated glucose uptake in muscle are not well defined and require further investigation. Another point to consider is that it has been suggested that the cellular distribution of DAGs, and ceramides, is more relevant to changes in insulin action than the total levels in tissue (*Perreault et al., 2018*; *Szendroedi et al., 2014*; *Chung et al., 2017*), but this was not investigated in our study.

In the liver, DAGs were increased in both the Hi-F and Hi-ST to a similar degree when compared to the chow controls and therefore the differences in insulin sensitivity observed in liver of Hi-ST and Hi-F mice are unlikely to be directly linked to differences in total DAG levels or differences in specific species (although the caveat of DAG cellular distribution still remains). However, ceramide species were different, particularly Cer22:0, which was one of the most abundant species. This particular species was increased in the Hi-F animals and was increased to a lesser extent in the Hi-ST, reflecting the intermediate protection against insulin resistance observed in the livers of the Hi-ST mice. Cer22:0 was also negatively correlated with the suppression of HGO, indicating that it may be involved in the development of insulin resistance. Previous studies have shown Cer22:0 to be increased in livers of people with type 2 diabetes (*Razak Hady et al., 2019*) and Hi-F fed rodents (*Turner et al., 2013*; *Montgomery et al., 2016*). Some studies investigating the role of ceramides in liver insulin resistance have implicated Cer16:0 as an important species (*Raichur et al., 2019*). In our study, this species was higher in the Hi-F animals but did not reach statistical significance. Our study clearly indicates that Hi-F diet-induced insulin resistance in muscle and liver is associated with increased DAG and ceramide levels, but that only lower levels of specific ceramides species correlate with the maintenance of insulin sensitivity, despite significant obesity and accumulation of tissue TAG in Hi-ST mice.

There has been some debate in recent times about the exact nature of the relationship between obesity and dysregulated glucose metabolism. While the majority of obese people exhibit some level of insulin resistance, there are obese individuals of both genders who have metabolic profiles indistinguishable from lean individuals (*Smith et al., 2019*; *Samocha-Bonet et al., 2012*). Whether these obese, insulin-sensitive people are on a trajectory to inevitable metabolic disease or have a genetic background or lifestyle that protects against metabolic disease, despite obesity, remains an area of

current research interest. One recent systematic review (*Vilela et al., 2021*) suggests that healthy eating patterns can influence the metabolic profile of otherwise obese individuals while other reviews suggest the variation in criteria used to classify obese metabolically healthy people make it difficult to draw conclusions about mechanisms for metabolic differences in similarly obese individuals (*Gómez-Zorita et al., 2021*). As reflected in our results in mice, it remains a possibility that the current diet may have a significant impact on insulin sensitivity irrespective of the obesity status of an individual. It is well documented that the availability of specific types of nutrients can alter glucose homeostasis in animals and humans. Acute infusion of lipids lowers insulin sensitivity within a few hours in humans and animals (*Dubé et al., 2014*; *Hoy et al., 2009*) and people undergoing routine screening with an oGTT are advised to consume largely carbohydrate foods for 24–48 hr before testing because of the effects of consuming fatty foods on subsequent glucose clearance (*Kaneko et al., 1998*). Together, this data suggests that a proportion of insulin action is adaptive and may be modulated by changing the diet over a relatively short period. This mechanism would enable organisms to conserve glucose for tissues unable to utilise fatty acids, such as the brain and central nervous system, during periods of carbohydrate scarcity.

Overall, our findings that mice fed a Hi-ST diet remain glucose tolerant and insulin sensitive despite having excess adipose tissue demonstrates that recent nutritional intake is an important factor in determining glucose homeostasis. Although the Hi-ST fed mice were not identical to lean control mice (they exhibited basal hyperglycaemia and hyperinsulinaemia), this dietary rodent model of insulin-sensitive obesity may be useful for investigating mechanisms involved in obesity-related insulin resistance. The differences in lipid composition in skeletal muscle and liver between Hi-ST and Hi-F mice also support the idea that individual lipid species may be more important determinants of insulin resistance than total lipid levels and therapeutic approaches that alter these specific lipids could have beneficial metabolic effects.

## Methods

All experimental procedures performed were approved by the Garvan Institute/St Vincent's Hospital Animal Ethics Committee and were in accordance with the National Health and Medical Research Council of Australia's guidelines on animal experimentation (protocol number 14_07).

### Animals

Male C57BL/6J$_{Arc}$ mice (8 weeks of age) were sourced from Animal Resource Centre (Perth, Australia) and after a 1 week acclimatisation period, mice were randomly assigned to receive either a standard chow diet (6% fat, 23% protein, and 71% carbohydrate, by calories; Gordon Specialty feeds, Sydney, Australia; energy density 13.0 kJ/g), a Hi-ST diet (22% protein, 57% carbohydrate, 21% fat, by calories), or a lard-based, Hi-F diet (22% protein, 21% carbohydrate, 57% fat, by calories), made in house and were available ad libitum. The exact dietary composition of the Hi-ST and Hi-F diets is shown in *Supplementary file 1c*. Importantly, water was added to both the Hi-ST and Hi-F formulated diets to increase palatability and diet was changed regularly (every 2 days). Mice were communally housed (four to five per cage) in temperature-controlled (29°C ± 0.5°C) and light-controlled (12 hr light:12 hr darkness cycle, 07:00–19:00 light) rooms with corn cob bedding.

### Assessment of body composition, respirometry, and energy intake

Lean and fat mass were measured using the EchoMRI-500 (EchoMRI LLC, Houston, TX, USA) according to the manufacturer's instructions, excluding body water. Whole body respirometry was performed on mice after 18 weeks of diet utilising the Promethion metabolic system (Sable Systems International, North Las Vegas, NV, USA). Mice were individually housed and acclimatised for 2 days in Promethion cages at 29°C. Food and water was available ad libitum and bedding was the same as in the home cage. The airflow of each chamber was 2 l/min and $O_2$ and $CO_2$ measurements were taken every 5 min across a 48–72 hr period and was averaged for every 40 min. Data were analysed by the ExpeData software package (Sable Systems International). Ambulatory activity in metres was taken from the 'all metres' readout and averaged for a 24 hr period. Energy intake measurements were performed on mice by the daily weighing of food hoppers and food spillage in communally housed cages and was averaged to account for multiple mice per cage. Food intake was corrected for the loss of water

by dehydration in the Hi-ST and Hi-F diets by measuring the rate of food weight loss over time in an empty animal cage.

## Glucose tolerance test

oGTT were carried out on 6 hr fasted mice at 2 pm (food removed at 8 am) after 4 and 12 weeks of dietary intervention. A fixed dose of 50 mg of glucose (200 µl of 25% glucose solution in water) was gavaged by laryngeal cannula. This dose corresponds to a dose of 2 g glucose/kg lean mass for a mouse with 25 g of lean mass. Blood glucose levels were monitored from the tail-tip using a hand-held glucometer (Accu-Check k Performa, Roche, Dee Why, Australia) before, and for 90 min following glucose administration. Insulin levels during the oGTT were measured in samples of whole blood collected from the tail using a mouse ultra-sensitive, ELISA kit (Crystal Chem, Elk Grove Village, IL, USA).

In one cohort of mice after 14 weeks of diet, a stable isotope labelled OGTT was performed as described previously (*Turner et al., 2018*; *Kowalski et al., 2015*). The administration of isotopically labelled glucose provides assessment of dynamic glucose disposal and pattern of EGP (*Bruce et al., 2021*). Glucose was measured with a glucose meter (Accu-Check, Roche, NSW, Australia) and plasma tracer enrichment measured by gas chromatography-mass spectrometry (*Turner et al., 2018*; *Kowalski et al., 2015*).

## Assessment of DNL

In one cohort of mice after 4 weeks of diet, the rate of DNL was measured. Briefly, at 8:00 in the postprandial state, mice were given an intraperitoneal injection of 0.5 mCi of $^3H_2O$ in 200 µl of sterile saline. One hour later, animals were killed by cervical dislocation and plasma, liver, WAT, and BAT were collected. Tissue saponification and determination of $^3H$ incorporation into fatty acids were conducted as previously described (*Cooney et al., 1989*). Rates of DNL were determined by dividing tissue lipid $^3H$ by the specific activity of $^3H$ in plasma correcting for time and tissue weight.

## Hyperinsulinaemic-euglycaemic clamps

Hyperinsulinaemic-euglycaemic clamps studies were performed as described in detail elsewhere (*Brandon et al., 2019*; *Brandon et al., 2016*) with slight modification. Briefly, after 20–26 weeks of diet, dual cannulation surgery was performed and at 6–8 days post surgery, clamps were performed with mice being held on a heat mat to maintain a warmer temperature. After a 90 min basal period, mice received a primed-continuous infusion of insulin (24 mU/kg bolus followed by 6 mU/kg/min; all animals were given the same dose based on a 30 g mouse in an attempt to get comparable insulin plasma levels) with euglycaemia maintained at ~8 mM during the clamp. A bolus of 10 µCi of 2 [$^{14}C$]-deoxyglucose was administered after euglycaemia was established for determination of tissue-specific glucose uptake. At the end of the tracer period (30 min), animals were euthanised and organs removed, snap-frozen in liquid nitrogen, and stored at –80°C for further analysis.

## Biochemical analysis

Plasma insulin, adiponectin, and leptin levels were determined by ELISA (mouse, ultra-sensitive, Crystal Chem, Elk Grove Village, IL, USA). Plasma non-esterified fatty acids (NEFA) were determined by NEFA kit (Wako Diagnostics, Mountain View, CA, USA). Tissue triglyceride content was measured in tissues as described previously (*Hoy et al., 2009*). Enzymatic assay of βHAD, CS, succinate dehydrogenase, and PDH was performed as previously described (*Small et al., 2019*; *Montgomery et al., 2013b*). Enzymatic activity of ACL and G6PDH was determined spectrophotometrically in tissue homogenates produced as previously described (*Small et al., 2019*) with the following reaction mixes: ACL, 100 mM Tris HCL, 10 mM MgCl$_2$, 10 mM DTT, 0.18 mM NADH, 20 mM sodium citrate, 0.3 mM CoA, excess malate dehydrogenase, pH 8.3. The reaction was initiated with 50 µl of 30 mM ATP. G6PDH, 100 mM Tris HCL, 0.5 mM EDTA, 0.02% BSA, 0.5 mM NADP, pH 9.4. The reaction was initiated with 50 µl of 12 mM glucose-6-phosphate.

## Immunoblotting

Immunoblotting was conducted as previously described (*Brandon et al., 2015*). Antibodies for Akt (#9272), pAkt (S473, #9271), ACC (#3662), FAS (#3180), SCD-1 (#2438), and cyclophilin (#2175) were

from Cell Signaling Technology (Danvers, MA, USA). Anti β-actin-HRP (#sc-47778 HRP) and anti-pan 14-3-3 (#sc-1657) were from Santa Cruz (Dallas, TX, USA). Anti-vinculin (#ab73412) antibody was from Abcam (Cambridge, UK). Protein content was determined by the Bradford assay and compared to a BSA standard.

## Histology

WAT was fixed in 10% formalin solution for 24 hr and then transferred to a 70% v/v ethanol solution. Fixed sections were embedded in paraffin blocks and transverse 4 µm sections were cut and stained with haematoxylin and eosin (H&E). Embedding, cutting, and staining were performed by the Garvan Institute Histopathology Core service. Pictures were taken using the Leica DMIL light microscope with an MC120 HD camera (Leica Microsystems, Wetzlar, Germany) at ×20 magnification. Pictures were processed using Leica application suite software (version 4). Adiposoft software (ImageJ, NIH, Bethesda, MD, USA https://imagej.net/Adiposoft) was used to measure adipocyte size.

## Tissue lipidomics

Lipids were extracted by a modified methyl-tert-butyl ether (MTBE) method as previously described (*Montgomery et al., 2013a*). Approximately 10 mg of frozen liver or quadriceps tissue were homogenised in methanol containing 0.01% butylated hydroxytoluene by bead homogeniser (Precellys 34 with Cryolys, Bertin Instruments, Montigny-le-Bretonneux, France). Samples were homogenised with 5 mm ceramic beads and samples were kept cooled to ≤4°C during homogenisation. The methanol used for homogenisation was spiked with a series of internal standards including: PC 19:0/19:0 (40 nmol), PE 17:0/17:0 (40 nmol), PS 17:0/17:0 (5 nmol), dihydrosphingomyelin 12:0 (5 nmol), ceramide 17:0 (1 nmol), DAG 17:0/17:0 (10 nmol), d5-triacylglycerol 16:0/16:0/16:0 (10 nmol), and CE 22:1 (2 nmol). Following homogenisation MTBE was added at a ratio of 3:1 (v/v), and samples were allowed to rotate overnight at 4°C. The following day 150 mM ammonium acetate was added to induce phase separation (1:3 v/v), samples were vigorously vortexed and then centrifuged for 10 min at 1000 × g. The upper phase containing lipids was transferred into a 2 ml glass vial and stored at –30°C prior to analysis. For sphingolipids analysis, a portion of the upper phases was taken and subjected to base hydrolysis using 10 M NaOH (rotated for 2 hr at room temperature). The same extraction procedure as described above was then used and the upper phase containing sphingolipids was transferred into a 2 ml glass vial and stored at –30°C until analysis.

Lipids were detected from extracts by shotgun lipidomics as described previously (*Norris et al., 2015*). Whole lipids extracts were diluted 350-fold in methanol:chloroform (2:1 v/v) containing 5 mM ammonium acetate and sphingolipid extracts were diluted 50-fold. Diluted samples were loaded onto a 96-well plate which was heat-sealed with foil and then centrifuged briefly (10 min, 2200 × g). Lipid extracts were analysed by direct infusion using an automated chip-based nanoelectrospray source (TriVersa Nanomate, Advion Biosciences, Ithaca, NY, USA) coupled to a hybrid triple quadrupole linear ion trap mass spectrometer (QTRAP 5500 Sciex, Framingham, MA, USA). Spray parameters were set at a gas pressure of 0.4 psi and a voltage of 1.2 and 1.1 kV for positive and negative ion mode respectively for all acquisitions. Lipid data were acquired using complementary precursor and neutral loss ion scans in positive and negative ionisation modes using settings previously described (*Norris et al., 2015*). Lipids were identified and quantified from internal standard using Lipidview Software (v1.2, Sciex, Framingham, MA, USA) as described previously (*Norris et al., 2015*).

## Tissue metabolomics

Glycolytic and TCA cycle metabolites were measured using negative mode ion-pairing liquid chromatography-mass spectrometry (LC-MS) method described previously (PMID 31690627). Briefly, 25 mg of powdered liver and muscle tissue was homogenised in 500 µl 50% (v/v) methanol:water mixture containing internal standards 2-morpholinoethanesulfonic acid and D-camphor-10-sulfonic acid at 2.5 µM. Equal volume of chloroform (500 µl) was added to the extracts and vortexed. The aqueous phase was separated from the insoluble and organic layers by centrifugation and was lyophilised using a Savant SpeedVac (Thermo Fisher Scientific). LC-MS analysis was performed using an Agilent Infinity 1260 LC coupled to an AB Sciex QTRAP 5500 MS. LC separation was achieved on a Synergi Hydro-RP column (Phenomenex, 100 mm length × 2.1 mm internal diameter, 2.5 µm particle size). Buffer A was 95:5 (v/v) water/acetonitrile containing 10 mm tributylamine and 15 mm acetic

acid (pH 4.9), and buffer B was 100% acetonitrile. MS source temperature set to 350°C, and 5 µl was injected. LC-MS data were extracted using MSconvert (version 3.0.18165-fd93202f5) and in-house MATLAB scripts. Molar concentrations were calculated using external standards injected alongside samples. This research was facilitated by access to Sydney Mass Spectrometry, a core research facility at the University of Sydney.

## Statistical analysis

Data are expressed as means ± SEM. Results were analysed by either a one-way ANOVA or two-way repeated measures ANOVA as appropriate. If the ANOVA reached statistical significance, a Tukey's post hoc test was used. Statistical analysis was performed in GraphPad Prism software (Prism 8, La Jolla, San Diego, CA, USA). Statistical significance was set at $p < 0.05$. Lipidomics data was analysed in R using a Kruskal-Wallis rank sum test with correction for false discovery rate using the method of Benjamini-Hochberg. Post hoc testing was conducted using pairwise Wilcoxon rank sum tests.

# Additional information

## Funding

| Funder | Grant reference number | Author |
|---|---|---|
| National Health and Medical Research Council | Fellowship 1003313 | Gregory J Cooney |
| National Health and Medical Research Council | Program grant 535921 | Gregory J Cooney |
| Diabetes Australia Research Trust | Grant | Gregory J Cooney |

The funders had no role in study design, data collection and interpretation, or the decision to submit the work for publication.

## Author contributions

Amanda E Brandon, Data curation, Formal analysis, Supervision, Methodology, Writing – original draft, Writing – review and editing; Lewin Small, Data curation, Formal analysis, Investigation, Methodology, Writing – review and editing; Tuong-Vi Nguyen, Eurwin Suryana, Henry Gong, Christian Yassmin, Sophie Stonehouse, Letisha Prescott, Data curation; Sarah E Hancock, Belinda Yau, Data curation, Formal analysis, Methodology; Tamara Pulpitel, Data curation, Formal analysis; Melkam A Kebede, Lake-Ee Quek, Greg M Kowalski, Clinton R Bruce, Data curation, Formal analysis, Methodology, Writing – review and editing; Nigel Turner, Formal analysis, Supervision, Methodology, Writing – review and editing; Gregory J Cooney, Conceptualization, Resources, Supervision, Funding acquisition, Investigation, Methodology, Project administration, Writing – review and editing

## Author ORCIDs

Amanda E Brandon http://orcid.org/0000-0002-4996-7189
Lewin Small http://orcid.org/0000-0002-9767-9464
Melkam A Kebede http://orcid.org/0000-0001-9686-7378
Clinton R Bruce http://orcid.org/0000-0002-0515-3343

## Ethics

All experimental procedures performed were approved by the Garvan Institute/St Vincent's Hospital Animal Ethics Committee and were in accordance with the National Health and Medical Research Council of Australia's guidelines on animal experimentation (protocol number 14_07).

## Decision letter and Author response

Decision letter https://doi.org/10.7554/eLife.79250.sa1
Author response https://doi.org/10.7554/eLife.79250.sa2

## Additional files

### Supplementary files

• Supplementary file 1. Tables showing (a) muscle metabolomics, (b) liver metabolomics, and (c) diet composition.

• MDAR checklist

• Source data 1. Raw data for *Supplementary file 1a*.

• Source data 2. Raw data for *Supplementary file 1b*.

### Data availability

All data generated or analysed during this study are included in the manuscript and supporting files; source data files have been provided for Figures 1,2,3,4,5, and 6.

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
