## [Editor Report]

This important study evaluates the effects of two distinct dietary methods that cause obesity in mice (high fat vs high starch) on insulin sensitivity and glucose homeostasis. The authors present compelling data showing that high starch feeding causes obesity without deleterious effects on insulin sensitivity. This work will have broad impact in the field and will help define the lipid mediators of metabolic disease.

---

## [Decision Letter]

**Decision letter after peer review:**

Thank you for submitting your article "Insulin sensitivity is preserved in mice made obese by feeding a high starch diet" for consideration by *eLife*. Your article has been reviewed by 3 peer reviewers, and the evaluation has been overseen by a Reviewing Editor and Mone Zaidi as the Senior Editor. The following individual involved in the review of your submission has agreed to reveal their identity: Will L Holland (Reviewer #3).

Essential revisions:

1) The authors need to address issues regarding body weight and composition, raised by Reviewer #1.

2) The authors need to address issues regarding adipose signaling and lipolysis, raised by Reviewer #2.

The reviewing editor and referees have agreed that addressing other issues would make the paper stronger, but is not essential for the publication of this work.

*Reviewer #1 (Recommendations for the authors):*

These studies tested whether obesity is always necessarily associated with insulin resistance or whether there are instances in which obesity does not promote insulin resistance. To this end, the authors fed mice either a low-fat chow diet, a high-fat diet HFD, or a high-starch diet (HSD). Although HFD and HSD-fed mice became equally obese, only HFD-fed mice displayed overt insulin resistance. HSD-fed mice displayed markers of increased de novo lipogenesis and carbohydrate handling, suggesting an improved ability to dispose of carbohydrates. Overall, these are important studies that add nuance to the common connection between obesity and insulin resistance.

Some issues that should be addressed are as follows:

1. Looking at the body weight and body composition data in Figure 1, there appears to be a significant discrepancy between the measurements of lean mass and fat mass, and total mass. Although body composition excluded total body water, there appears to be a ~5g difference between total body weight and the sum of fat and lean mass in the obese mice (both HFD and HSD). The authors should explain this difference and whether it can be accounted for by body water.

2. The following statement on page 11, lines 212-214 could use some modification: "Additionally, this data suggests that in diet-induced obese mice, hepatic insulin resistance seems to be the largest contributing factor to GIR during a clamp". This may be true, but the data can only support this statement only under the conditions used in these studies. Based on the description of the clamp in the methods, insulin infusion rates were calculated based on a 30g mouse. Although the rationale for this is clear and justified, this may lead to an unintended consequence that impacts the interpretation of the data. The insulin levels achieved for the chow-fed mice will be adequate, but 30g is significantly smaller than the body weight in HFD and HSD-fed mice. Thus, these mice were exposed to lower insulin levels relative to their body weight. Since the liver is more sensitive to insulin than are non-hepatic tissues, lower relative insulin levels are likely to result in a bias towards hepatic phenotypes. It is possible that if insulin infusions had been calculated based on body weight, differences in glucose disappearance may have been observed. This should be taken into consideration for the statement above.

3. It would be interesting to determine whether insulin clearance and levels of incretin hormones are different between HFD and HSD-fed mice if the authors still have sufficient plasma samples from the oGTTs to measure C-peptide and GLP-1 levels.

*Reviewer #2 (Recommendations for the authors):*

This was a nicely performed study that will be important in the field. Too commonly, mouse models of obesity are used without the complete characterization of their effects on insulin sensitivity with the assumption that obesity always leads to insulin resistance.

Given the well-published connection between adipose tissue insulin resistance and lipolysis to HGP, the authors should add additional experiments from the clamp studies to solids the effects of insulin action on white fat. Circulating levels of glycerol and FFAs should be reported. Additionally, canonical insulin signaling via AKT in white adipose tissue should be reported. These studies will help define the underlying mechanisms driving the increase in HGP following the HFD diet and would be important to add to the body of literature suggesting lipolysis is a key determinant of liver insulin sensitivity. Moreover, it would be important to know if insulin signaling in white adipose is also unaffected like liver and muscle or if white fat behaves differently. Understanding this would also allow for proper interpretation of Figure 4, which documents increased adipose tissue weight and size despite decreased glucose uptake (Figure 3).

---

## [Author Response]

Reviewer #1 (Recommendations for the authors):Some issues that should be addressed are as follows:1. Looking at the body weight and body composition data in Figure 1, there appears to be a significant discrepancy between the measurements of lean mass and fat mass, and total mass. Although body composition excluded total body water, there appears to be a ~5g difference between total body weight and the sum of fat and lean mass in the obese mice (both HFD and HSD). The authors should explain this difference and whether it can be accounted for by body water.

Not all animals that were weighed also had their body composition measured (body weight n=18-20 across the 3 experimental groups; body composition n=8-10) and there may be small differences in measured mass between cohorts of mice. Looking at the animals that were both weighed and had their body composition assessed, the fat mass and lean mass accounts for approximately 90% of the body mass and total free body water could account for the rest. Body water was excluded from the body composition analysis as described in the methods (line 472) because it significantly increases the time required for analysis.

2. The following statement on page 11, lines 212-214 could use some modification: "Additionally, this data suggests that in diet-induced obese mice, hepatic insulin resistance seems to be the largest contributing factor to GIR during a clamp". This may be true, but the data can only support this statement only under the conditions used in these studies. Based on the description of the clamp in the methods, insulin infusion rates were calculated based on a 30g mouse. Although the rationale for this is clear and justified, this may lead to an unintended consequence that impacts the interpretation of the data. The insulin levels achieved for the chow-fed mice will be adequate, but 30g is significantly smaller than the body weight in HFD and HSD-fed mice. Thus, these mice were exposed to lower insulin levels relative to their body weight. Since the liver is more sensitive to insulin than are non-hepatic tissues, lower relative insulin levels are likely to result in a bias towards hepatic phenotypes. It is possible that if insulin infusions had been calculated based on body weight, differences in glucose disappearance may have been observed. This should be taken into consideration for the statement above.

In order to interpret hyperinsulinemic-euglycemic clamp data, steady state plasma insulin levels must be similar between animals (irrespective of the dose and rate of insulin infusion). Insulin action in all tissues will be determined by the circulating availability of insulin and not by the plasma level relative to body weight. We achieved this (Figure 3B) by infusing a standard dose of insulin as basal levels of insulin were already elevated in the Hi-F and Hi-ST groups which we knew from the GTT data and from previous experiments clamping obese mice/rats. We agree with the reviewer that clamps are not physiological conditions and the circulating level of insulin achieved in the clamp setting (~150uU/ml) was higher than seen in any of the groups in the oGTT (~80uU/ml in the Hi-F and Hi-ST), which is likely to be a more physiological response. This is why we believe that our statement is correct as we very specifically say that hepatic insulin resistance seems to be the largest contributing factor to GIR during a clamp and not in the physiological setting. However, we have modified this sentence so that it now reads:

“Additionally, this data suggests that in diet-induced obese mice, hepatic insulin resistance seems to be the largest contributing factor to the lower GIR during a clamp, while during a GTT, hepatic and peripheral insulin resistance seem to make similar contributions to the relative glucose intolerance observed in Hi-F mice (Figure 2 I,J).”

3. It would be interesting to determine whether insulin clearance and levels of incretin hormones are different between HFD and HSD-fed mice if the authors still have sufficient plasma samples from the oGTTs to measure C-peptide and GLP-1 levels.

We agree that GLP-1 and c-peptide would be interesting to measure, unfortunately we do not have any blood/plasma from the GTT. We are ethically limited in the volume and number of blood samples we can take during an oGTT and insulin was considered a priority. We will keep this in mind for future experiments.

Reviewer #2 (Recommendations for the authors):This was a nicely performed study that will be important in the field. Too commonly, mouse models of obesity are used without the complete characterization of their effects on insulin sensitivity with the assumption that obesity always leads to insulin resistance.Given the well-published connection between adipose tissue insulin resistance and lipolysis to HGP, the authors should add additional experiments from the clamp studies to solids the effects of insulin action on white fat. Circulating levels of glycerol and FFAs should be reported. Additionally, canonical insulin signaling via AKT in white adipose tissue should be reported. These studies will help define the underlying mechanisms driving the increase in HGP following the HFD diet and would be important to add to the body of literature suggesting lipolysis is a key determinant of liver insulin sensitivity. Moreover, it would be important to know if insulin signaling in white adipose is also unaffected like liver and muscle or if white fat behaves differently. Understanding this would also allow for proper interpretation of Figure 4, which documents increased adipose tissue weight and size despite decreased glucose uptake (Figure 3).

On page 9, we do say that:

“All 3 groups had similar basal NEFA and were able to suppress NEFA levels to a similar extent (data not shown).”

In line with your comments, we have placed the values in the text. Unfortunately, we do not have any basal plasma samples left to do the glycerol determination.

Due to the similar NEFA suppression, similar adipocyte hypertrophy as well as similar insulin resistance seen in adipose depots of both dietary intervention groups, we believe the differences in metabolism of white adipose tissue is not playing a particular role in the differences in whole-body, muscle or hepatic insulin resistance observed in the different diet groups. This is the reason why we did not pursue additional biochemical or signalling measurements in white adipose. However, in line with the reviewers request, we have inserted Akt phosphorylation for the eWAT and sWAT into figure 3 which also shows no difference in signalling between the groups (Figure 3L and M).